

# Functions of 1,25-dihydroxy vitamin D3, vitamin D3 receptor and interleukin-22 involved in pathogenesis of gout arthritis through altering metabolic pattern and inflammatory responses

Yuqi Chen[1], Huiya Ma[2], Youwei Du[3], Jianjian Dong[1], Chenkai Jin[1], Lihui Tan[1] and Rong Wei[1]

[1] Department of Rheumatology and Immunology, the People's Hospital of Soochow New District, SuZhou, China
[2] College of Chemistry and Pharmacy, Northwest A&F University, Yangling, China
[3] State Key Laboratory of Crop Stress Biology for Arid Areas and College of Plant Protection, Northwest A&F University, Yangling, China

Corresponding author
Rong Wei, 952257684@qq.com

## ABSTRACT

**Background.** Gouty arthritis (GA) is a common type of inflammatory arthritis. Recent studies demonstrated that 1,25-dihydroxy vitamin D3 (1,25(OH) 2 VD3) and vitamin D3 receptor (VD-R) play a protective role in acute inflammation, but interleukin-22(IL-22) promotes inflammation, especially for arthritis. However, our understanding of the responses of 1,25(OH) 2VD3 and IL-22 to gout was still unclear. Presently, in-depth metabolomics, bioinformatics and clinical characteristics analyses were performed to elucidate the pathogenesis and valuable clinical indicators of gouty arthritis.

**Methods.** Peripheral venous blood was taken for investigation. The levels of IL-22 and $1,25(OH)_2VD3$ were determined in patient's plasma via ELISA, and the mRNA levels of IL-22 and VD-R were measured via qRT-PCR. The interaction network of VD-R and IL22 were constructed by the Search Tool for the Retrieval of Interacting Genes/Proteins (STRING), and the biological function of the related proteins were analyzed by Clusterprofiler Metabolomics were performed to decipher the metabolic variations of GA.

**Results.** The levels of VD-R and 1,25(OH) 2 VD3 were identified to be low. What' s more, GA patients were reported to have high expression of IL-22. And IL-22 levels positively correlated with C-reactiveprotein (CRP) serum levels in the bivariate correlation analysis, whereas the level of 1,25(OH) 2VD3 negatively correlated with that of CRP. GO and KEGG analyses revealed that IL-22 and 1,25(OH) 2 VD3 were involved in stress immunity and inflammatory responses. These pathways are known to play a role in GA pathogenesis. Meanwhile, the metabolic profiles of GA serum showed that the increase in various amino acids and uric acid are involved in GA pathogenesis. Importantly, VD-R and IL22 closely correlated with the level of key metabolites uric acid, whose increase promoted the occurrence of GA.

**Conclusion.** GA patients have low levels of VD-R and 1,25(OH) 2 VD3, and high levels of IL-22 together with various amino acids and uric acid. The levels of IL-22 and 1,25(OH) 2VD3 were positively and negatively correlated with C-reactive protein
(CRP) serum levels, respectively. Both IL-22 and 1,25(OH) 2 VD3 functioned in GA-related immune and inflammatory responses, and closely correlated with the level of GA-related uric acid. Overall, IL-22, VD-R and 1,25(OH) 2 VD3 play functionally important roles in inflammatory responses and are relevant to gout pathogenesis.

## INTRODUCTION

Gout is a major public health problem impacting human health (*Stack et al., 2019*). Currently, the prevalence of gout was higher ranging from 1–4%, and patients with gout suffer from dramatic joint inflammation and excruciating arthritic pain, which profoundly reduced the patients' life quality (*Singh & Gaffo, 2020*; *Yin et al., 2020*). Meanwhile, comorbidities such as hypertension, stroke, and heart disease, chronic kidney disease are common in people with gout and complicate its management and disease outcomes (*Kuwabara et al., 2017*). It is a complex disorder caused by the deposition of monosodium urate (MSU) crystals in joints and soft tissues (*Zhu, Deng & Zhou, 2018*). MSU stimulates the activation of the innate immune system and subsequently releases an array of inflammatory mediators (*Nakayama, 2018*; *Yin et al., 2020*). These inflammatory factors trigger a strong inflammatory response in joints and periarticular tissues (*Richette & Bardin, 2010*). However, the exact pathogenesis of gout is yet to be elucidated.

Elevated levels of specific pro-inflammatory cytokines are known to trigger highly reactive inflammatory infections. Intriguingly, 1,25(OH) 2 VD3 can inhibit inflammation and is commonly found in a range of immune cells. This suggests it plays a key role in the immune system (*Ho et al., 2020*). Adding to this hypothesis, Wen H indicated that 1,25(OH) 2 VD3 may play as an immune modulating agent to suppress Th17 cell cytokines, such as IL-22 (*Wen et al., 2015*). In this study, we hypothesize that IL-22, is correlated with the occurrence and development of gout. We also hypothesized that 1,25(OH) 2 VD3 plays an anti-inflammatory role in gouty arthritis and can even suppress episode of gout.

This study investigated changes in the serum levels of 1,25(OH) 2 VD3 and IL-22 in the peripheral blood of gout arthritis patients. Correlations between serum levels and sedimentation rates of CRP, serum uric acid, creatinine, triglycerides, cholesterol and platelets with 1,25(OH) 2 VD3, and IL-22 were subsequently investigated. Finally, the biological functions of 1,25(OH) 2 VD3 and IL-22 in gout pathogenesis were further explored *via* bioinformatics analysis, and the detailed variations in metabolic pattern of GA were displayed by metabolomics profiles. These findings provide valuable insight into the mechanisms underpinning gout pathogenesis and provide more ideas for the treatment of gouty arthritis.

## METHODS

### Gout patients recruitment

A total of 58 gout outpatients were recruited from the Rheumatism and Immunology Department of the People's Hospital of Suzhou New District, China. These patients were recruited between March 2019 and September 2019. To be included in the study, patients had to fit into the criteria defined by the American Rheumatism Association (now named the American College of Rheumatology (ACR)) in 1977 (*Wallace et al., 1977*) and the ACR/European League Against Rheumatism (EULAR) in 2015 (*Neogi et al., 2015*). Patients were categorized into two groups: acute (in the last week, with joint swelling and pain, and increased white blood cell or erythrocyte sedimentation rate, n = 30) and non-acute (no joint swelling and pain occurred in the last month, acute joint swelling and pain occurred in the past, and no colchicine, prednisone, non-steroidal anti-inflammatory drugs and urate-lowering therapy was given, n = 28) GA. Patients with secondary gout as a result of drugs, chemoradiotherapy, malignant tumor, endocrine and kidney diseases were excluded from the study. Twenty healthy physical examining persons were recruited as healthy controls who were lacking hyperuricemia and metabolic syndromes as well as other chronic diseases. Participants provided fasting morning venous blood samples and the specific operation was completed by nurses. Participants were matched for age and sex. Written informed consent was obtained from all patients at the time of admission. The People's Hospital of Suzhou New District granted Ethical approval to carry out the study within its facilities (Ethical Application Ref: 2018-006).

### Laboratory indexes

Laboratory indexes were determined in the laboratory of Suzhou High-Tech Zone at the People's Hospital. Blood cell analysis was performed with the blood cell analyzer LH750 (Beckman, Brea, CA, USA). Analyses including erythrocyte sedimentation rate (ESR), CRP, uric acid, urea, and creatinine were performed with a DxC800 automatic biochemical analyzer (Beckman).

### Quantitative PCR (qPCR)

Whole peripheral blood samples were used for total cellular RNA isolation. Total RNA was extracted using a standard TRIZOL method following (*Qin et al., 2013*) and cDNA was generated using the PrimeScript RT Reagent kit (Takara). IL-22 and VD-R expression was quantified *via* qPCR using a LightCycler 480 and SYBR Green I Master Mix (Roche). Expression levels were normalized to glyceraldehyde 3-phosphate dehydrogenase (GAPDH). Water was used as a negative control lacking template. A standard curve was generated for each assay. Amplification efficiency was calculated based on the mRNA standard curve. Assays were performed in duplicate. The average used in statistical analyses.

### Enzyme-linked immunosorbent assay (ELISA)

Peripheral blood was collected in heparin anticoagulant tubes. Plasma was obtained *via* centrifugation at 3000 RPM for 15 min. Prior to examination, plasma was stored at –20 °C. Plasma levels of IL-22 and 1,25(OH)2VD3 were determined using a quantitative sandwich

technique, for which ELISA kits (eBioscience, San Diego, CA, USA) were used according to the manufacturer's instructions.

## Enrichment and protein–protein interaction network analysis

Gene ontology (GO) enrichment analysis of differentially expressed genes was performed using the ClusterProfiler R package. GO terms with a corrected $p > 0.05$ were considered enriched by differentially expressed genes. The Kyoto Encyclopedia of Genes and Genomes (KEGG) database was subsequently used. This database contains a wealth of understanding of molecular-level information, including large-scale molecular datasets generated *via* genome sequencing and other high throughput technologies (http://www.genome.jp/kegg/).

## Metabolomics analysis

To investigate the potential variations in the metabolisms of patients, the serum of GA patients and healthy controls were collected for a pilot of metabolomics analysis. To identified the most relevant metabolic changes involved in the pathogenesis of GA, we used pooled serum extracts from five replicated individuals as one sample. Detailed steps of metabolomics analysis are as follows: $50 \pm 1$ ml sample from GA patients ($N = 15$) and healthy control ($N = 15$) were extracted with pre-cold extraction mixture (methanol/chloroform (v:v) = 3:1) with 10 μL internal standard (adonitol), and samples were extracted with ultrasonication for 5 min in ice water. Every five extracts were mixed into one pooled sample. In total, three pooled replicates of GA group or healthy group were used for the pilot metabolomics. After evaporation in a vacuum concentrator, 40 μL of Methoxyamination hydrochloride (20 mg/mL in pyridine) was added and then incubated at 80 °C, then derivatized by 60 μL of BSTFA regent (1% TMCS, v/v) at 70 °C for 1.5 h.

All samples were then analyzed by gas chromatograph using an Agilent 7890 with a time-of-flight mass spectrometer (GC-TOF-MS) and DB-5MS capillary column. 1 μL aliquot of sample was injected in splitless mode. Helium was used as the carrier gas with 1 mL min$^{-1}$ flow rate. The energy was $-70$ eV in electron impact mode. The mass spectrometry data were acquired in full-scan mode with the m/z range of 50–500 at a rate of 12.5 spectra per second.

## Raw data preprocessing

Raw data analysis was finished with Chroma TOF (V 4.3x, LECO) software and LECO-Fiehn Rtx5 database was used for metabolite identification with an in-house database. The noise and deviation values were removed by filtering individual peaks and the interquartile range, respectively. Only the peak area data with a hollow value not more than 50% in all groups were retained, then the missing value was simulated by 1/2 of the minimum value. And the data were normalized by an internal standard (IS). The data variation were performed by R packages XCMS software. The Principle Component Analysis (PCA) and pathway enrichment analysis were performed by Metaboanalyst 3.0. The PCA showed the metabolic variations of GA patients. PCA was performed on the three-dimensional metabolic data involving the metabolite name, sample name and normalized peak area. The data were further treated through mean centering and unit variance scaling. The PCA plots was

generated to interpret cluster separation. And the therapy $P < 0.05$ and FC > 2 were used for identification of significantly differential metabolites.

## Statistical analysis

Statistical analyses were performed using SPSS Statistics Version 21 (IBM). The groups were tested using two-sided $t$ test or ANOVA for continuous variables and the Chi-squared statistic for categorical variables. A two-sided $p < 0.05$ was used to denote statistical significance. Baseline characteristics were presented as mean (with an associated standard deviation (SD)) for continuous variables. For dichotomous variables, proportions (%) were used. Mann–Whitney U tests were used for non-normally distributed variables. A Chi-square test or Fisher's exact test were used for categorical variables. Pearson or Spearman's rank correlation tests were used depending on the data distribution.

## RESULTS

### Patient characteristics

The clinical characteristics of recruited gout patients were documented and analyzed. These characteristics included age, body mass index (BMI), uric acid and creatinine levels, platelet levels, erythrocyte sedimentation rate (ESR) and C-reactive protein (CRP) levels, cholesterin levels, triglyceride levels, blood glucose levels, and tophi deposits. Other comorbidities were also documented. A summary of the patient characteristics is shown in Table 1. In total, the study recruited 30 AGA patients, 28 NAGA patients, and 20 healthy controls. Participants were age- and sex-matched. It's important to note that the BMIs of patients in the AGA and NAGA groups were significantly higher than those in HCs ($p < 0.05$). This supports the notion that obesity may be a risk factor for gout. Besides, Creatinine levels are higher in patients with gout than in normal subjects, reflecting the fact that gout can impair kidney function. What's more, UA, blood glucose, and ESR were all significantly higher in AG patients than in HC patients. In addition, in AGA patients, CRP levels were found to be highly upregulated in comparison to the NAGA and HC groups. There were no differences in the disease duration, hypertension rates, and diabetes rates in the NAGA patients *vs.* the AGA patients ($p > 0.05$) (Table 1).

### IL-22, VD-R and 1,25(OH)$_2$ VD3 levels closely correlated with gout pathogenesis

IL-22 is a highly important cytokine and performs various anti-inflammatory and pro-inflammatory functions *in vivo*. In this study, qRT-PCR was performed to investigate the expression levels of IL-22. In comparison to the HC group, IL-22 expression was highly upregulated in the AGA and NAGA groups. On another note, as the receptor for vitamin D, VD-R plays a key role in regulation of the immune system (*Medrano et al., 2018*). Remarkably, this study reported that VD-R expression levels were dramatically downregulated in the AGA and NAGA groups (Fig. 1). Subsequent, ELISA analysis and qRT-PCR experimentation revealed a consensus expression level of IL-22 and VD3. What's more, ELISA indicated that IL-22 protein expression was significantly upregulated in both the AGA and NAGA groups. In addition, 1,25(OH)$_2$VD3 protein expression was also

**Table 1  Patient characteristics.**

| Variable | AGA Patients N = 30 | NAGA Patients N = 28 | HC N = 20 | p |
|---|---|---|---|---|
| Age, y | 43 ± 15.68 | 36.7 ± 11.4 | 43.0 ± 8.6 | |
| BMI kg/m$^2$*# | 26.6 ± 3.5 | 27.4 ± 4.5 | 24.23 ± 1.3 | *=0.022, #=0.003 |
| UA umol/L* | 446.63 ± 150.95 | 419.14 ± 149.20 | 341.45 ± 66.99 | *=0.08 |
| Cr umol/L*# | 100.72 ± 26.33 | 92.58 ± 19.34 | 79.72 ± 13.19 | *=0.001, #=0.041 |
| Platelet 10$^9$/L | 218.83 ± 63.97 | 250.43 ± 71.45 | 227.15 ± 54.68 | |
| ESR mm/h | 20.2 ± 5.2 | 13.60 ± 4.8 | 10.8 ± 3.5 | |
| CRP mg/L* Δ | 24.42 ± 13.42 | 2.45 ± 1.50 | 1.85 ± 1.68 | *< 0.001, Δ<0.001 |
| Cholesterin mmol/L | 4.30 ± 0.83 | 4.57 ± 0.98 | 4.23 ± 0.6 | |
| Triglyceride mmol/L | 1.97 ± 0.96 | 2.45 ± 1.5 | 1.85 ± 1.68 | |
| Blood glucose mmol/L* | 5.64 ± 1.06 | 5.35 ± 0.48 | 5.03 ± 0.51 | *< 0.011 |
| Tophi n | 5 (16.7%) | 3 (10.7%) | Not available | |
| Disease duration m | 27.8 ± 34.5 | 41.9 ± 52.5 | Not available | |
| *Comorbidities* | | | | |
| Hypertension | 7(24.1%) | 7 (25.0%) | 1 (5%) | |
| Diabetes | 4(13.3%) | 5 (17.9%) | 0 (0.0%) | |

**Notes.**

AGA, acute gouty; BMI, body mass index; CRP, C-reactive protein; ESR, erythrocyte sedimentation rate; HC, healthy controls; NAGA, non-acute gouty arthritis; UA, uric acid.

\*$p < 0.05$, between AGA patients *vs.* HC.

#$p < 0.05$, between NAGA patients *vs.* HC.

Δ$p < 0.05$, between NAGA patients *vs.* AGA patients.

identified to be significantly decreased in the AGA group by ELISA assay. There was no clear difference between the NAGA and HC group in terms of 1,25(OH)$_2$VD3 protein expression levels (Fig. 1).

## Levels of IL-22 and VD-R closely correlated with GA-related metabolites

In order to further identified the potential metabolic variations of GA patients, untargeted metabolomics was performed. This pilot of metabolomics analysis was performed on the serum from both HC and GA patients. Three replicates were performed for each group. Principal components analysis (PCA) was performed on the metabolic profiles of both the HC and GA patients. This analysis showed that both groups had a distinct profile. This showed that there is measurable variation in metabolic profiles of HC and GA patients (Fig. 2C). Furthermore, we constructed a coexpression network on IL22, VDR and metabolites from metabolomics. We found various metabolites involved in Aminoacyl-tRNA biosynthesis, Pantothenate and CoA biosynthesis, Purine metabolism, Phenylalanine, tyrosine and tryptophan biosynthesis, Arginine biosynthesis, Valine, leucine and isoleucine biosynthesis, beta-Alanine metabolism and Phenylalanine metabolism were closely correlated with IL22 and VDR (Fig. 2A; Table 2). We noted that six metabolites including 3-Hydroxyanthranilic acid, Palatinose, sulfuric acid, serine, 2,6-Diaminopimelic acid, Phytol and uric acid were positively correlated with IL22, whereas their correlation with VDR were negative (Fig. 2A). The level of these metabolites were higher in GA patients

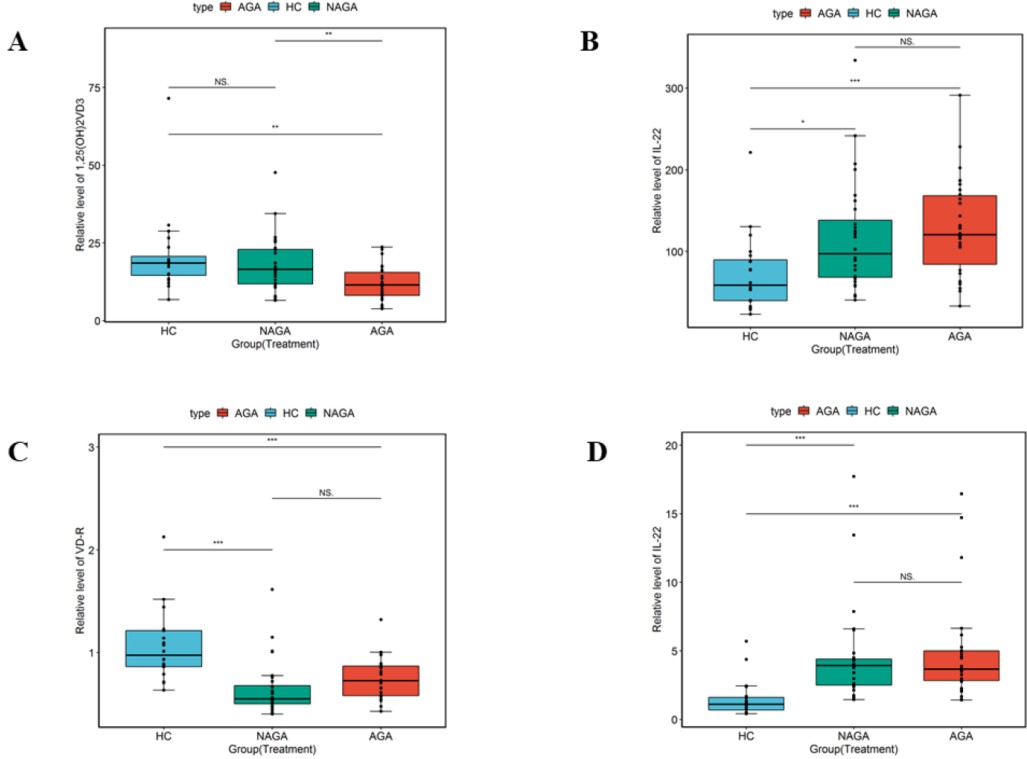

**Figure 1** **MRNA and protein levels of IL-22, VD-R and 1,25 (OH)₂VD3 in the plasma from gouty arthritis patients.** (A) Serum 1,25(OH)2VD3 levels in HC, AGA, and NAGA groups (mean ± SD), $p$-value < 0.01; (B) Serum IL-22 levels in HC, AGA, and NAGA groups (mean ± SD), $p$-value < 0.01; VD-R (C) and IL-22 (D) mRNA levels in serum, Data represent mean ± SD. $p$-value < 0.01;

than that in HC (Fig. 2B). Of note, metabolic alterations performed various functions in the pathogenesis of GA (*Hoque et al., 2020*). Subsequently, 446 metabolites were identified in the serum. Out of these metabolites, the levels of 147 metabolites were highly increased in serum of GA patients. The levels of 33 metabolites were decreased in the serum of GA patients (Fig. 2D). Subsequent pathway enrichment analysis, using the KEGG database, was performed on the 147 metabolites upregulated in GA patients. This analysis identified 45 pathways, nine pathways of which were significantly enriched (Fig. 2E). This included the following pathways: aminoacyl-tRNA biosynthesis; arginine biosynthesis; pantothenate and CoA biosynthesis; alanine, aspartate and glutamate metabolism; phenylalanine, tyrosine and tryptophan biosynthesis; glycine, serine and threonine metabolism; galactose metabolism; glutathione metabolism and valine; leucine and isoleucine biosynthesis, revealing the influence of these metabolic alterations which correlated with IL22 and VDR in the pathogenesis of GA. In this study, a variety of amino acid-related pathways were significantly activated in GA patients. This indicates that a high level of amino acids are involved in GA pathogenesis. Notably, research has shown that elevated levels of proteinuria is indicative of arthritis (*Luo et al., 2018*; *Teumer et al., 2019*). In agreement, this study identified that serum uric acid levels of GA patients were 2.8-fold higher than that

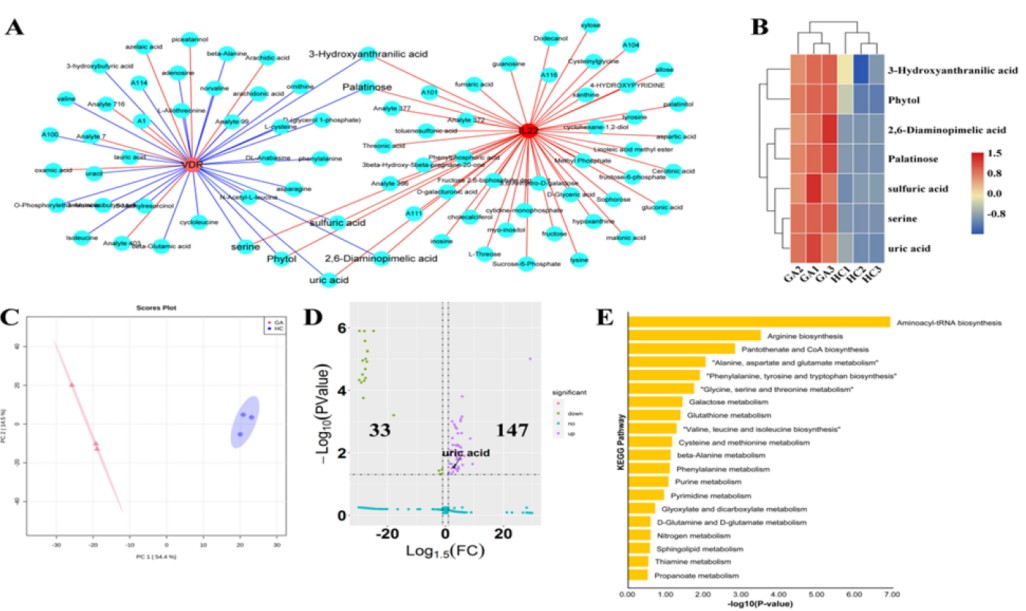

**Figure 2  Metabolomic landscape of the GA patients.** (A) Coexpression network between IL22, VDR and metabolic signature; Negative and positive relationship were shown represented by red and blue line, respectively; (B) Relative level of metabolites which were closely associated with both IL22 and VDR; (C) PCA plots of the metabolomics patterns of the GA and HC groups (D) Volcano plots displayed differential metabolites in the metabolomics data of the GA and HC groups (E) Pathway enrichment analysis of the upregulated metabolites of GA patients.

**Table 2  Pathway analysis on metabolites closely associated with IL22 and VDR.**

| Name | Hits | Raw p | Impact |
|---|---|---|---|
| Aminoacyl-tRNA biosynthesis | 9 | 1.64E−05 | 0.16667 |
| Pantothenate and CoA biosynthesis | 5 | 0.000285 | 0.02143 |
| Purine metabolism | 7 | 0.004921 | 0.03226 |
| Phenylalanine, tyrosine and tryptophan biosynthesis | 2 | 0.006345 | 1 |
| Arginine biosynthesis | 3 | 0.009984 | 0.06091 |
| Valine, leucine and isoleucine biosynthesis | 2 | 0.027162 | 0 |
| beta-Alanine metabolism | 3 | 0.030917 | 0.39925 |
| Phenylalanine metabolism | 2 | 0.04182 | 0.35714 |

in the HC group (Fig. 2B). Uric acid is known for its inductive function in pathogenesis of GA (*Huang et al., 2020*). Remarkably, levels of IL-22 and VD-R closely correlated with the uric acid levels in the serum of both HC and GA patients. Thus, We concluded that IL22 and VD-R were closely correlated with the GA-related metabolic alterations, supporting their functions in the pathogenesis of GA.

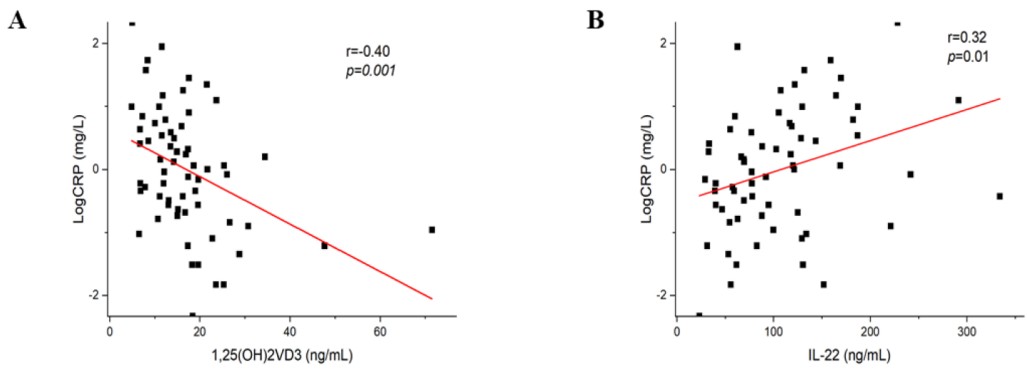

**Figure 3** (A–B) Correlations in the serum levels of LogCRP and IL-22 or 1,25(OH)$_2$VD3.

## IL-22 and 1,25 (OH)$_2$ VD3 levels in serum closely correlated with CRP levels

As shown in Fig. 3, the level of IL-22 in the serum positively correlated with CRP levels ($r = 0.32, p < 0.05$). In addition, 1,25(OH)$_2$VD3 levels negatively correlated with CRP levels ($r = -0.40, p < 0.01$) (Fig. 3). CRP is a non-specific inflammatory marker. Interestingly, IL-22 levels in the serum positively correlated with CRP levels. However, the serum levels of 1,25(OH)$_2$D3 negatively correlated with the serum CRP levels. Correlation analysis was performed to confirm the correlations between the levels of IL-22, 1,25(OH)$_2$D3 and other measurements such as platelets, ESR. No significant correlations were identified between these indicators. Similarly, the correlation between CRP and IL-22 and VD3 are not similar for all 3 groups of participants. These findings suggested that 1,25(OH)$_2$VD3 and IL-22 are associated with inflammatory expression in gout arthritis.

## IL-22 and VD-R are imperative in the inflammatory response network

In order to further investigate the function of IL-22 and VD-R in the pathogenesis of gout, a complex protein-protein interaction network was constructed using Search Tool for the STRING. The created network showed that various genes were involved in inflammatory response. These genes included STA3, MED25, MED22, STAT1, CREB1, HIF1, and IL-10 (Fig. 4D). These interactions indicated that IL-22 and VD-R could in fact further influence protein functions associated with the inflammatory response. The topology of this complex network was subsequently analyzed. This analysis showed that IL-22 and VD-R were at the core of the network. This suggests important roles for both IL-22 and VD-R in the regulation inflammatory responses (Fig. 4D).

Furthermore, GO enrichment analysis was performed on the identified regulatory network. This analysis showed that these proteins were involved in various terms associated with the cellular component category. This included various components: transcription factor complex, RNA polymerase II transcription factor complex, protein-containing complex, nucleus, nucleoplasm part, and nuclear transcription factor complex (Fig. 4A). For the molecular function category, protein functions were mainly involved in vitamin D receptor binding, transcription regulator activity, transcription factor binding, thyroid

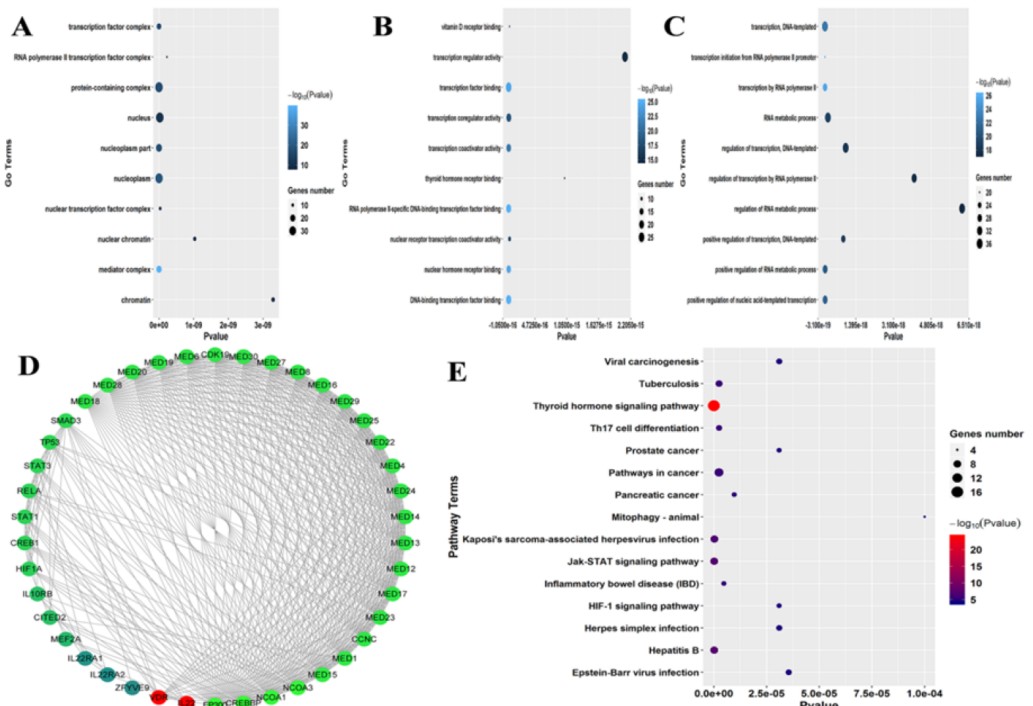

**Figure 4** **Complex regulation network and imperative roles of IL-22 and VD-R were involved in GA-related inflammatory responses.** (A) Cellular component analysis on the regulatory network of IL-22 and VD-R; (B) molecular function analysis on the regulatory network of IL-22 and VD-R; (C) biological progress analysis on the regulatory network of IL-22 and VD-R; (D) the regulatory network of IL-22 and VD-R; (E) pathway enrichment analysis on VD-R and IL-22.

hormone receptor binding, and RNA polymerase II-specific DNA-binding transcription factor binding (Fig. 4B). It was also noted that various terms associated with biological progress were overrepresented. This included transcription, DNA-templates, transcription initiation from the RNA polymerase II promoter, RNA metabolic process, regulation of transcription, and positive regulation of nucleic acid-templated transcription (Fig. 4C). All these terms closely correlated with the transcription activity *in vivo* (Fig. 4).

Subsequently, pathway enrichment analysis was performed on these target genes using the KEGG database. Pathway enrichment analysis revealed that various pathways associated with inflammatory responses system were significantly enriched. This included: viral carcinogenesis, tuberculosis, rhyroid hormone signaling pathway, prostate cancer, pathways in cancer, JAK/STAT signaling pathway, inflammatory bowel disease (IBD), HIF-1 signaling pathway, herpes simplex infection, hepatitis B, and Epstein-Barr virus infection (Fig. 4E). In summary, these results agreed that IL-22 and VD-R play functionally important roles in the inflammatory response and are therefore relevant to gout pathogenesis (Fig. 4).

## DISCUSSION

Gout arthritis is an auto-inflammatory disease (*Salehzadeh et al., 2019*). The acute onset of gout is an inflammatory process induced by urate crystals. These crystals manifest as

inflammatory arthritis and cause painful disability. Various researchers have proposed that the gout arthritis is closely linked with various inflammatory reactions, including NLRP3, IL-1 and IL-22 (*Cabau et al., 2020*; *Luo et al., 2017*). However, the role of IL-22 and VD3 in gout pathogenesis is yet to be elucidated. In this study, GO and KEGG pathway analyses identified important roles of both IL-22 and $1,25(OH)_2VD3$ in stress immunity as well as inflammatory responses. These processes are known to be associated with gout arthritis pathogenesis. Remarkably, the levels of IL-22 were shown to be higher in patients with gouty arthritis in either the acute phase or non-acute phase in comparison to healthy controls. What's more, decreased levels of VD-R mRNA and $1,25(OH)_2$ VD3 in both AGA and NAGA groups were verified *via* ELISA and qRT-PCR. Furthermore, the levels of IL-22 and VD3 correlated with that of CRP in gout patients. Importantly, we found that uric acid highly accumulated in GA patients, and the levels of VD-R and IL22 closely correlated with the level of uric acid, supporting the imperative roles of VD-R and IL22 in the pathogenesis of GA. In summary, this study reported the involvement of IL-22 and VD3 in inflammation associated with gout pathogenesis. These results suggest a pivotal role of IL-22 and VD3 in mediating gout inflammation. These findings provide a novel treatment strategy for gout.

IL-22 is the source of increasing attention in a wide range of inflammatory and autoimmune diseases due to its specific function *in vivo*. Remarkably, this study showed that mRNA expression levels of IL-22 were higher in the acute and remission phases of gout. This indicates that IL-22 was involved in the pathogenesis of gouty arthritis. Previous research has proven the role of IL-22 in the regulation of inflammatory responses associated with inflammatory diseases by binding to the IL-22 receptor. This receptor is transmembrane in nature, and is composed of two subunits (IL-22R1 and IL-10R2) (*Lu et al., 2015*); Perusina (*Lanfranca et al., 2016*). Notably, IL-22 is characterized as a two-faced cytokine. This means it performs not only deleterious roles but also protective roles (*Huan et al., 2016*). It is also an important driver of autoimmune diseases and is involved in the pathogenesis of arthritis (*Yang & Zheng, 2014*). A range of studies have shown that patients with rheumatoid arthritis and psoriatic arthritis frequently have abnormal IL-22 expression (*Ezeonyeji et al., 2017*; *Miyazaki et al., 2018*). IL-22 promoted inflammatory response and osteoclast formation in rheumatoid arthritis *via* the promotion of fibroblast-like synoviocytes proliferation and monocyte chemoattractant protein (MCP) expression (*Li et al., 2019*). *Marijnissen et al. (2011)* reported IL-22 and IL-22 receptor levels to be increased in synovial tissue during inflammation. On another note, IL-22 promoted the expression of IFN-γ, thereby inhibiting intestinal coronavirus replication (*Xue et al., 2017*). IL-22 and IFN-γ have been shown to suppress pulmonary inflammation in pneumonia caused by *Pseudomonas aeruginosa* (*Broquet et al., 2020*). The therapeutic benefits of IL-22 pathway activation have been demonstrated in experimental colitis (*Lamas et al., 2016*). In this study, levels of IL-22 in plasma were increased in both acute and chronic gout patients. These levels correlated positively with CRP levels. This suggests the imperative role of IL-22 in the pathogenesis of gout arthritis.

As an active vitamin D *in vivo*, $1,25(OH)_2VD3$ plays role in maintaining the balance of calcium and phosphorus various biological processes by binding to VD-R. Detailed studies have shown that $1,25(OH)_2VD3$ plays an important role in immune regulation,

anti-inflammation, and anti-infection (*Holick, 2014*). In this study, we reported that the levels of VD-R mRNA and protein were significantly decreased in both the AGA and NAGA group. This suggests that VD-R may play a negative regulatory role in the inflammatory process of gout arthritis. The decrease in 1,25(OH)$_2$VD3 levels in serum are potentially caused by high uric acid levels (*Chen et al., 2014*). A study has described that 1,25(OH)$_2$VD3 effectively inhibited Th17 cell differentiation *in vitro* and regulated the immune response of T cells. The study reported that 1,25(OH)$_2$VD3 treatment could cause decreases in the levels of IL-17A, IL-22 and IFN-γ (*Gu, Xu & Cao, 2017*). What's more, 1,25(OH)$_2$VD3 promoted the formation of STAT1-VD-R complexes in monocytes. Interestingly, 1,25(OH)$_2$VD3 significantly reduced the transcriptional activity of VD-R, but enhanced the transcription of STAT1. Thus, it can be hypothesized that correlations between the VD-R and STAT signaling pathways exist, and that the anti-inflammatory effects of 1,25(OH)$_2$VD3 may be related to the effects of vitamin D on the JAK/STAT pathway. Another study found that 1,25(OH)$_2$VD3 can in fact control the expression of VD-R, signal transducers, and transcriptional activator 5 (STAT5) *via* the regulation of signal transduction pathways (*Yang et al., 2015*). KEGG pathway analyses showed that VD-R and IL-22 mRNA were mainly concentrated in multiple signal transduction pathways such as the JAK/STAT signaling pathway in tuberculosis, prostate cancer, and inflammatory bowel disease. VD-R can combine with STAT3 to inhibit the JAK/STAT signaling transduction pathway (*Zhang et al., 2020*). Research has also proven that 1,25(OH)$_2$ D3 can be expressed *via* the promotion of STAT5 and its phosphorylation (*Gu, Xu & Cao, 2017*). This can in turn promote Th17 cells specific cytokine IL-17 as well as the expression and secretion of IL-22. Andoh et al. reported that IL-22 raised IL-6, IL-8, IL-11, and other inflammatory mediators and chemokine expression by activating SEMF nf-kappa B cells, JAK-2/STAT and AP-1-3 signaling pathways (*Andoh et al., 2005*). Thus exerting a pro-inflammatory effect.

As a non-specific inflammatory marker, CRP can bind to a variety of eukaryotic and prokaryotic pathogens, facilitating complement activation through the classical pathway (*Pepys & Hirschfield, 2003*). This leads to immune activation, lymphocyte infiltration, immune molecules consumption, and widespread inflammation. Levels of IL-22 positively correlated with serum CRP levels, which confirmed that IL-22 was involved in the inflammatory response in the pathogenesis of gout arthritis. Serum 1,25(OH)2D3 levels negatively correlated with serum CRP levels. This indicated that VD3 played a protective role in gout. In addition, functional enrichment and KEGG signaling pathway analyses of target genes predicted the molecular regulatory networks related to disease occurrence. In parallel, we detailedly depicted the landscape of metabolic alterations between gout patients and normal human that the levels of various amino acids and uric acid associated with the gout pathogenesis were higher in gout patients. What's more, serum VD-R and IL-22 were increased in patients and may be involved in the pathogenesis of gout arthritis *via* influencing the uric acid level of gout patients that both VD-R and IL-22 closely correlated with the level of uric acid of GA patients. This goes some way to elucidate the molecular regulatory network controlling gout arthritis pathogenesis.

However, it is important to note that there are limitations to the interpretation of these results due to the small sample size. And the metabolomics analyses is also a pilot

of concept study. Thus, some conclusions relevant to metabolic variations of GA patients were hypothetical at this stage. However, our study provided multiple evidences that the levels of 1,25(OH)$_2$VD3 and IL-22 were reported to be abnormal in the serum of patients with gout arthritis, and both of them involved in the pathogenesis of gout arthritis. And the specific mechanism underpinning this remain unclear. Therefore, further studies are required to confirm the mechanisms *via* which 1,25 (OH)$_2$VD$_3$ affects the secretion of IL-22 in gout.

## ACKNOWLEDGEMENTS

Thanks are due to D. He for valuable discussion.

### Funding

This work was supported by Key Project of Scientific Innovation Fund Grant of the People's Hospital of Suzhou New District (SGY2018C02). The funders had no role in study design, data collection and analysis, decision to publish, or preparation of the manuscript.

### Grant Disclosures

The following grant information was disclosed by the authors:
Key Project of Scientific Innovation Fund Grant of the People's Hospital of Suzhou New District: SGY2018C02.

### Competing Interests

The authors declare there are no competing interests.

### Author Contributions

- Yuqi Chen, Huiya Ma and Youwei Du conceived and designed the experiments, performed the experiments, analyzed the data, prepared figures and/or tables, authored or reviewed drafts of the paper, and approved the final draft.
- Jianjian Dong performed the experiments, analyzed the data, prepared figures and/or tables, and approved the final draft.
- Chenkai Jin and Lihui Tan performed the experiments, prepared figures and/or tables, and approved the final draft.
- Rong Wei conceived and designed the experiments, analyzed the data, prepared figures and/or tables, authored or reviewed drafts of the paper, and approved the final draft.

### Human Ethics

The following information was supplied relating to ethical approvals (i.e., approving body and any reference numbers):

The People's Hospital of Suzhou New District granted Ethical approval to carry out the study within its facilities (Ethical Application Ref: 2018-006).

## Data Availability

The raw measurements are available in the Supplementary Files.

## Supplemental Information

Supplemental information for this article can be found online at http://dx.doi.org/10.7717/peerj.12585#supplemental-information.

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
