# Peer review of "Functions of 1,25-dihydroxy vitamin D3, vitamin D3 receptor and interleukin-22 involved in pathogenesis of gout arthritis through altering metabolic pattern and inflammatory responses"

_PeerJ, doi:10.7717/peerj.12585_

## Round 0.1 · original submission · Major Revisions

Please note in particular the suggestions of the first reviewer and to address all of these, as well as those of the second reviewer.

Reviewer 1 ·

Basic reporting

This study is very valuable to the medical field as well as to the general public. The contribution of Vit D and its metabolites is becoming increasingly important in disease control.

Major issues:
• Although the reasoning behind the study is clear after reading the complete paper, the problem statement is not clearly formulated in the introduction section. For example, it is stated that gout is a problem in human health, but not how much (epidemiology) and how (symptoms). It is mentioned that MSU deposits, but not where, and that gout is triggered by fluctuations in uric acid, but the reason why this happens is not mentioned. Apart from the ending sentence, to provide more insight into the disease, how could this study contribute to medicine? Could it perhaps improve treatment? Enhance diagnosis?

• I do feel that the writing style can be improved. For example, may sentences begin with the word ‘And’. Section 2.1 line 66 ‘In’ should be ‘in’. Abbreviations are not defined upon first mention (section 2.2 line 79 for example), and some, such as KEGG and TB are not defined at all.

• Writing is not always clear, for example, Introduction line 45 – cytokines trigger inflammation, but is inflammation a known symptom of gout? Line 45 – Vit D inhibit inflammation, which form/metabolite of Vit D?

• Minor issues:
• Change the ‘vitamin receptor’ keyword to Vitamin D receptor’
• I would suggest that full descriptions, and not abbreviations, are used in the title of the paper. These abbreviations might not be standard and known to the whole scientific community.
• Section 2.2 line 78 delete ‘what’s more’
• Line 273 – Pseudomonas aeruginosa should be in italics.

Experimental design

Major issues:

• Patient information can be described in more detail (section 2.1) – were all patients and controls hospitalised? Where were controls recruited? In which country is this Hospital? It is mentioned in line 63 that the criteria had to be met – it will add content to the section if these criteria could be mentioned here. Patient received ‘regular’ treatment – does this refer to ‘normal’ or ‘chronic’ treatment? It will be good if it is also mentioned here that controls and patients were aged and sex matched. Did controls also provide written consent? How were samples collected – by a doctor in hospital, were patients fasting, what time of day, etc.

• No detail is provided on the quantification, zero-replacement, normalisation and cantering of the untargeted GC-MS data. Also no information on the similarity hit for annotation of compounds or signal to-noise-ratio used.

• The PCA and methods for identification of important metabolites from the untargeted data are not described in the methods section.

Minor issues:

• Is TRIzol a kit?

• It is mentioned that PCR was done in duplicate, was the average used in statistical analyses?

• Add G-force and time to centrifugation steps.

• It is stated that 1 mg sample was used, should this perhaps be 1 mL?

• It is not clear if the proteins were removed after the cold extraction, before the evaporation.

• Line 119 – ‘a two sided p<0.05’ please add the test, ANOVA or t-test.

• Baseline characteristics are presented as mean or median – I would suggest using the same value.

Validity of the findings

Major

• It is concerning that only 3 samples from 2 groups were used for untargeted analyses. The rationale behind this is not clear. Metabolomics power calculations would indicate that these groups are not statistically viable. Conclusions made can therefore only be speculations. In addition, all 3 groups of samples should have been included to make concrete conclusions. This is of major concern, and I suggest that GC-MS analyses are done on all samples in the cohort, or as an alternative, the importance of the metabolomics data and the conclusions made should be made less prominent, and describe only as a pilot study.

• It is not clear how the important metabolites were identified, based on which criteria. The reason for doing pathway analyses twice, with different sets of metabolites, is also unclear.

• It should be mentioned whether or not the correlations between CRP and IL-22 and VD3 are similar for all 3 groups of participants.

Minor:


• In line 132 it is mentioned that any other co-morbidities were documented, although only diabetes and hypertension are noted – were these the only co-morbidities?
• Please define STRING

·

Basic reporting

The manuscript has been well planned and is in the main clearly written and understandable. The authors are commended for their ability to report the vast amount of data generated effectively in a relatively succinct manner. The language employed is generally of good quality with only a few colloquial errors in the manuscript.

Referencing in the main is of a high standard. The authors employ references well throughout the manuscript. The one area where referencing may appear more weak is in the introduction. The references employed are acceptable. I however, feel that more references here alongside the explanation for why the study was conducted would be valuable.

The figures and tables used within the manuscript are of high quality. The normal conventions has been adhered to and thus the comprehensibility of the figures are high. In general the structure of the manuscript and figures contribute highly to the good readability thereof.

Experimental design

The experimental design of this study is based on a well identified knowledge gap. The methodology employed is acceptable in the main, including high technical and ethical standards being employed. There is some discomfort however, in the fact that the authors did not appear to have performed any sort of sample size calculation, nor any calculation of the statistical power of their findings. This is highly regrettable, as the findings appear very promising. The repeatability thereof, along with the generalizability thereof is however, in doubt due to the small sample size. The discussion of methods and results is good, but the conclusion should reflect on the fact that this may only serve as a preliminary study, based on the small sample size.

Validity of the findings

The authors have discussed findings well and in line with the data provided. All bench and calculated data has been provided in a clear and usable manner. The only criticism remains the small sample size that is not justified or discussed in the manuscript.

Additional comments

This is to my mind an excellent piece of work. The only detracting factor, being the sample size. In all other aspects I was impressed with the high standards achieved by the authors.

---

## Round 0.2 · Minor Revisions

Thanks for the revisions, just a few suggestions remain to improve the document.

Reviewer 1 ·

Basic reporting

The reviewer is happy with the changes made.

Experimental design

I still feel that this is just a pilot/proof of concept study in terms of the metabolomics analyses, and although it was added and the end of the manuscript that the limitation is the small sample size, it should also be mentioned in the methodology section that the metabolomics section is only a pilot study and that the conclusions made are just hypothetically at this stage. With high throughput systems such as the Leco GC-TOFMS, 58 samples can easily be analyzed and aligned using the statistical compare function of ChromaTOF. It should also be added in the methodology that pooled samples were used, since, at present, it is indicated that only 3 patient samples were include per group, and the pooling is not mentioned at all. Also, the reasoning behind analyzing aliquots of a pooled sample as opposed to using individual patient samples should explicitly be mentioned. The use of a pooled sample is not advisable for disease metabolomics, since, in practice, this lowers your sample cohort to 1 sample per group and it does not include individual variation at all.

PCA is still not described in the methods section, but the PCA result is discussed in the discussion section.

Validity of the findings

The reviewer is happy with the changes

·

Basic reporting

Upon reviewing the amended manuscript, it is clear that the authors accepted the review comments from both reviewers and have done their best to comply with the suggestions of the reviewers. This is highly appreciated. It appears that shortcomings in the original manuscript, pertaining mostly to a limited introductions, some lack of description in the methods and some acknowledgement of the small sample size, has all been appropriately addressed by the authors. I am pleased to report, that with this being considered, the manuscript now appears suitable for publication in my opinion. The use of language and literature is of a acceptable quality. The methodology used and the data is available and clear. The conclusion is supported by the data and contributes to the knowledge in the field.

Experimental design

The experimental design is in my opinion rigorous and as I have indicated is now more clear and understandable. The data analysis is now also more clearly explained.

Validity of the findings

Findings are meaningful and well stated. The only concern is the relatively small sample, size, which authors acknowledge. The findings are still meaningful in my opinion and may lead to further research.

---

## Round 0.3 · accepted · Accept

Thanks for addressing these final requests.

Reviewer 1 ·

Basic reporting

The reviewer is happy with the changes made

Experimental design

The reviewer is happy with the changes made

Validity of the findings

The reviewer is happy with the changes made

Additional comments

The reviewer is happy with the changes made